# Experimental Study on the Mechanical Properties of Crumb Rubber Concrete after Elevated Temperature

**DOI:** 10.3390/polym15143102

**Published:** 2023-07-20

**Authors:** Yang Han, Zhishuan Lv, Yaqiang Bai, Guoqi Han, Dongqiao Li

**Affiliations:** 1College of Civil Engineering, Kashi University, Kashi 844008, China; ksdxlzs@163.com (Z.L.); 2210577@tongji.edu.cn (G.H.); 2Institute of Engineering Disaster Prevention and Mitigation, Henan University of Technology, Zhengzhou 450001, China; 3Zhengzhou University Mult-Functional Design and Research Academy Co., Ltd., Zhengzhou 450002, China; baiyaqiang127@163.com; 4School of Civil Engineering, Tianjin University, Tianjin 300354, China; dongqiao2008@tju.edu.cn

**Keywords:** crumb rubber, concrete, elevated temperature, mechanical properties

## Abstract

To reduce the environmental damage caused by waste rubber, crumb rubber concrete (CRC) was prepared by replacing some fine aggregates with crumb rubber. The effects of elevated temperature as well as crumb rubber content on the mechanical properties of the prepared CRC were studied. The crumb rubber contents were 0%, 10%, and 20%, while CRC was subjected to atmospheric temperatures (AT) of 300 °C, 500 °C, and 700 °C. The concrete without crumb rubber content was used as the control group at the atmospheric temperature. The mass loss, thermal conductivity characteristics, compressive strength, splitting tensile strength, axial compressive strength, elastic modulus, and stress-strain characteristics of CRC at elevated temperatures were studied. The experimental results show that: (1) With the increase in crumb rubber content and temperature, the cracks on the surface of the specimen gradually widen while the mass loss of the specimen increases. (2) With the increase in crumb rubber content and temperature, the cube compressive strength, splitting tensile strength, axial compressive strength, and elastic modulus of CRC decrease, yet the plastic failure characteristics of CRC are more obvious. (3) The influences of elevated temperature on strength and elastic modulus are as follows: splitting tensile strength > elastic modulus > axial compressive strength > cubic compressive strength. (4) With the increase in temperature, the stress-strain curve of the CRC tends to flatten, the peak stress decreases, and the corresponding peak strain significantly increases. With the increase in crumb rubber content, there is a great decrease in peak stress, yet the corresponding peak strain is basically the same. The use of CRC can be prioritized in applications that increase toughness rather than strength.

## 1. Introduction

In the 21st century, humanity has entered the circular economy era, and the concept of a “circular society” has become established in countries around the world as an important path toward sustainable development [1,2]. With the rapid development of the automotive industry, the number of waste rubber tires has sharply increased. In 2022, 13.3 billion rubber tires were produced annually in China, and more than 50 million waste rubber tires were added annually, equivalent to approximately 1.8 million tons. Currently, incineration and landfilling are the simplest and cheapest methods for treating waste rubber. However, once rubber is burned, carbon dioxide is emitted, toxic gases such as sulfur dioxide, nitrogen oxides, and carcinogenic dioxins are released, and the natural environment is severely polluted [3,4,5]. Rubber is difficult to decompose under natural conditions, and soil becomes severely polluted and loses its vitality through landfilling [6,7]. Due to the difficulty of handling them, most waste tires are not recycled, resulting in serious environmental pollution, as shown in Figure 1. Dealing with this “black pollution” has become a global challenge for both the environment and resources. Finding a way to recycle and make harmless these waste rubber resources is of great significance [8].

Since the 1980s, many scholars have been adding waste tire rubber powder to concrete to prepare CRC [9,10,11,12]. The addition of crumb rubber not only improves upon the disadvantages of the heavy weight and high brittleness of concrete but also expands the application field of waste rubber recycling. CRC is a composite based on ordinary concrete that is formed by adding finer crumb rubber to replace some of the fine aggregate or larger crumb rubber to replace some of the coarse aggregate. Research has shown that CRC outperforms ordinary concrete in terms of energy absorption and consumption [13,14], fatigue resistance [15], reducing material brittle failure [16], deformation capacity [17], hindering crack development [18], durability [19,20,21], and thermal performance [22]. To reduce the degree of brittle failure of deep foundation pit support structures, the application of CRC to deep foundation pit support has been studied [16]. The addition of rubber powder has a positive impact on the durability of concrete and can effectively reduce the migration rate of chloride ions in the concrete [21]. However, with the addition of crumb rubber, the elastic modulus, compressive strength, and tensile strength of CRC decrease to varying degrees [23,24,25]. By adding mineral admixtures such as slag [26], the pore structure of CRC can be optimized, and mechanical properties such as compressive strength can be improved. The compressive strength and impact resistance of CRC can also be improved by preheating and hardening the surface of crumb rubber [27]. In addition, in order to make full use of the denaturation ability of CRC, steel fiber has recently been introduced by some scholars to enhance the impact toughness of concrete [28]. CRC has currently been applied to crumb rubber concrete blocks [29], CRC bridge panels [30], self-compacting concrete [31], permeable concrete [32,33], rubber asphalt concrete pavement [34], and flexible components [16].

Although concrete has good fire resistance, its mechanical properties may deteriorate under elevated temperatures, leading to explosive spalling and irreparable damage to the structure, which endangers the overall safety of the structure [35,36,37]. Therefore, it is necessary to study the fire resistance performance of concrete [38,39,40]. Appropriate fire safety requirements specified in the building code for concrete structural elements must be met. The behavior of a concrete structural member exposed to fire is dependent on concrete properties, etc., including thermal, mechanical, and deformation [35]. Crumb rubber is a polymer mainly composed of rubber and carbon black. Crumb rubber generally softens at around 200 °C and burns at around 350 °C. The crumb rubber will be integrated into the concrete after degradation, which will have a certain impact on the concrete [41]. At present, most of the research on CRC performance has focused on atmospheric temperature, with limited research on performance under elevated temperatures. After elevated temperatures, the changes in the internal structure are more complex in CRC than in ordinary concrete, with differences also in the strength loss and failure mechanism [42,43]. In addition, the influence of elevated temperature on the mechanical properties of CRC needs to be compared, which can provide a certain reference for the application of CRC. In order to study the internal fracture failure of CRC, the acoustic emission method and SEM technique have been used by some scholars to observe the internal failure of concrete and expand the index of additional characterization ability of the mechanical properties of CRC after elevated temperatures [44,45]. Therefore, analysis of the failure mechanism and residual strength of CRC after elevated temperature is of great significance, and such analysis can provide a reference basis for the fire resistance design, fire resistance limit analysis, and residual bearing capacity calculation of CRC structures after fire. In this paper, the effects of crumb rubber content and heating temperatures on the strength, elastic modulus, and stress-strain of CRC were studied, as shown in Figure 2.

## 2. Materials and Methods

### 2.1. Materials and Preparation of the CRC

The cement is a 42.5-grade ordinary Portland cement produced by China Sunstone Group, and the physical and mechanical performance indicators of the cement are shown in Table 1. Fly ash is of high-quality Grade II. Polycarboxylic acid, a high-efficiency water-reducing agent, is adopted. The fine aggregate is natural Zone II sand produced in Lushan, Henan, China, with a fineness modulus of 2.9. The coarse aggregate is produced in Xingyang, Henan, China, and is limestone crushed stone with a particle size of 5–20 mm and continuous grading. The crumb rubber is black tire rubber particles produced by Jiaozuo Rubber Factory in Henan, China, with particle sizes of 0.6 mm, 1–3 mm, and 3–5 mm. Through preliminary experimental research, it was found that among the three sizes of crumb rubber, crumb rubber of 3–5 mm particle size has the least impact on the compressive and tensile properties of natural sand concrete. Therefore, a particle size of 3–5 mm was selected as the crumb rubber used in this experiment, with an apparent density of 1080 kg/m^3^, as shown in Figure 3. The grading curve of the aggregate is shown in Figure 4.

As a nonpolar material, crumb rubber has weak bonding ability with cementitious materials. In addition, there are acidic substances on the surface of waste rubber that affect the bonding between crumb rubber and cement gel. In order to improve the bonding performance between crumb rubber and cementitious materials, the crumb rubber is immersed in a 3% NaOH alkaline solution to remove the acidic substances on the rubber surface, exposing the polar bonds inside the rubber that closely combine with the cement gel [46]. The modifier is added step by step, with half of the NaOH solution poured into the crumb rubber and continuously stirred. After 15 min, the remaining modifier is poured in and stirred for another 15 min. This is allowed to stand for 12 h, then rinsed with clean water multiple times until the crumb rubber surface is free of any detectable greasiness. Furthermore, it is laid flat and dried for later use.

CRC is prepared based on C30 concrete, using an equal volume of crumb rubber to replace fine aggregates with substitution rates of 0%, 10%, and 20%, respectively. The specimens are RC (reference concrete), CRC (crumb rubber concrete)-10, and CRC-20, as shown in Table 2.

To ensure uniform mixing of CRC, the preparation process is as follows: (1) The crumb rubber and fine aggregate are first mixed dry in a mixer for 5 min at a speed of 60 r/min, and the crumb rubber is evenly dispersed in the fine aggregate. (2) According to the mix proportion in Table 2, cement, fly ash, and coarse aggregate are sequentially added to the mixer and stirred for 3 min at a speed of 60 r/min. (3) The water and water-reducing agent are evenly mixed, poured into the mixer, and stirred for 3 min at a speed of 90 r/min. The prepared CRC is added to the test mold in two steps, with approximately the same thickness each time. Manual vibration with two steps is used to prevent the crumb rubber from floating during the vibration process and ensure the consistency of the test results. In the first vibration, the tamping rod is inserted multiple times in a spiral direction from the edge to the center to eliminate air inside the material. In the second vibration, the tamping rod should remain vertical, penetrate the upper layer, insert 20–30 mm into the lower layer, and repeat the first vibration. The surface of the test mold is covered with plastic film. After 24 h, the mold is removed, and the test piece is placed in a constant temperature and humidity curing box with a temperature of 20 ± 1 °C and a relative humidity of 95% for 28 days.

### 2.2. Experimental Design

Experience and experiments with fire have shown that the strength loss of concrete is relatively small before the action of 300 °C, and after exceeding 700 °C, the strength is basically lost. Therefore, in this experiment, three target temperatures of 300 °C, 500 °C, and 700 °C were selected to heat the concrete at elevated temperatures. Atmospheric temperature was used as a control group, and tests for individually determining the cube compressive strength, splitting tensile strength, uniaxial compressive strength, elastic modulus, and uniaxial stress-strain relationship of the CRC after elevated temperature were carried out. The number, size, grouping, and effective number of test pieces are shown in Table 3. The cured specimen is allowed to stand in a dry environment for another 10 days and then placed in a high-temperature furnace for heating at a rate of 15 °C/min. After reaching the preset temperature, it is kept at a constant temperature for another 90 min. The door of the high-temperature furnace is then opened, and the high-temperature furnace and the specimen are completely cooled to atmospheric temperature. The specimen is taken out and left to stand for 2 days before conducting mechanical tests.

The KSL-30-12YSM track-type double door rapid heating high-temperature furnace was used in the elevated temperature test, as shown in Figure 5. The three-phase 380 V power supply is adopted, with a maximum heating rate of 30 °C/min and a maximum heating temperature of 1250 °C. Three nickel cadmium nickel silicon type thermocouples are placed in different parts of the furnace to control and monitor the temperature inside the furnace, maintaining a consistent temperature within the furnace with a temperature control accuracy of ±2 °C.

To ensure the representativeness of the measured temperature, a thermocouple is buried every 50 mm along one end face of the specimen at the center of the cross-section, as shown in Figure 6. To ensure uniform heating of the specimen in the elevated temperature furnace, the specimen is placed in two layers on a pre-made iron frame, and thermocouples are placed at different positions to monitor the surface and internal temperature of the test piece, as shown in Figure 7.

The microcomputer hydraulic universal testing machine produced by Shenzhen Xinsansi Company was used for loading, with the model CHT4106 and a maximum applied force of 1000 kN. The loading rates of cube compressive strength and splitting tensile strength are 0.3–0.35 MPa/s and 0.05–0.08 MPa/s, respectively. The TM-2 concrete elastic modulus tester produced by Yangzhou Hengtian Technology Development Co., Ltd. (Yang zhou, China). was used for deformation measurement.

## 3. Results and Discussion

### 3.1. Elevated Temperature Test Phenomenon

When the test lasts for 15 min, the furnace temperature reaches 250 °C, and water vapor and a pungent odor overflow from the furnace door. At 25 min, the furnace temperature reaches 400 °C, and a large amount of water vapor overflows from the furnace door, as shown in Figure 8. At 90 min, the water vapor and smoke near the furnace door gradually dissipate, but there is still the pungent smell of burning rubber. During the elevated temperature test, none of the specimens showed any explosive spalling phenomena. However, the edges and corners of the specimen show varying degrees of loose and local spalling. The spalling of the specimen corner becomes more obvious with an increase in temperature and crumb rubber content.

At 300 °C, the surface of the specimen shows a reddish-brown color. The surface of the specimen mixed with crumb rubber shows holes of varying sizes and obvious microcracks, while the control group specimen does not show any cracks. At 500 °C, the specimen turns light gray, and some CRC specimens show obvious defects at the edges and corners. There are obvious holes on the surface of the specimens, and some of the holes still contain incomplete burnt crumb rubber residues. Cracks are formed between the holes, creating a clear sense of looseness. The control group specimens show subtle cracks. At 700 °C, the specimen turns gray-white. The edges and corners of the CRC specimen can be easily crushed by hand, resulting in a more pronounced sense of looseness and severe strength loss. The crumb rubber on the surface of the specimen has completely disappeared, leaving only white holes. Most of the surface cracks are interconnected, with these holes as the centers, as shown in Figure 9. With the increases in crumb rubber content and temperature, the number of cracks gradually increases, and the cracks gradually widen.

### 3.2. Quality Loss of the CRC

After an elevated temperature, with the loss of free and bound water, the combustion or melting of crumb rubber, and a series of physicochemical reactions inside the concrete, the mass of the specimen is lost to varying degrees, as shown in Figure 10. With the increase in temperature, the mass loss of the specimen increases. With the increase in crumb rubber content, the mass loss is more obvious.

### 3.3. Thermal Conductivity Characteristics of the CRC

Figure 11 shows the time-temperature curves at different positions on the surface and inside of the specimen at 700 °C. The thermal conductivity of rubber is about 0.2 W/m°C, making it a poor conductor of heat. The rise in temperature at the center of the specimen was much slower than the surface and furnace temperatures, and this is due to the low thermal conductivity of the concrete [47]. As shown in Figure 11, with the addition of crumb rubber, the thermal conductivity of concrete decreases, and with the increase in crumb rubber content, the decrease in thermal conductivity is more pronounced. On the one hand, the reduction in thermal conductivity of CRC is because of the change in internal moisture. On the other hand, the thermal conductivity of crumb rubber is lower than that of concrete.

Different temperature measurement points inside the specimen exhibit a temperature plateau at 100 °C, which has been discussed in previous literature [48,49]. When the temperature of several measuring points inside the concrete members reaches about 100 °C, there is an obvious temperature stagnation period. The free water in the specimen is extensively evaporated at about 100 °C, taking away a large amount of heat during the evaporation process, resulting in a brief temperature plateau in the time-temperature curve. The surface temperature of CRC is higher than that of the control group, while the internal temperature of CRC is lower than that of the control group. There are two possible reasons for this phenomenon. On the one hand, crumb rubber on the surface of CRC was melted and burned. On the other hand, due to the lower thermal conductivity of CRC, heat conduction to the interior was slower, which may lead to heat accumulation on the surface.

During the heating stage, a significant temperature gradient is formed inside the specimen. The closer it is to the heating surface, the faster its heating rate and temperature after reaching the constant temperature stage. During the constant temperature stage, the temperature inside the specimen continues to rise until the end of the constant temperature stage, and the temperature inside the specimen does not reach the set furnace temperature, indicating that the concrete has very obvious thermal inertia. During the cooling stage, the closer the specimen is to the heating surface, the faster the temperature drops, while the central position of the specimen remains at a relatively high temperature until the end of temperature collection.

### 3.4. Cube Compressive Strength

At the initial loading stage of the specimen at atmospheric temperature, there was no significant change on the surface of the control group specimen. With the increase in load, subtle vertical cracks appear on the surface of the specimen, and the load continues to increase, rapidly expanding, leading to brittle failure. CRC exhibits obvious plastic failure characteristics. At the initial stage of loading, similar to the control group, there is no significant change on the surface. Approaching the failure load, subtle cracks appear on the surface of the specimen. The load continues to increase, the number of cracks increases, and they gradually widen, ultimately losing bearing capacity but remaining relatively intact. The larger the crumb rubber content, the more complete the specimen becomes after failure, as shown in Figure 12. After elevated temperatures, the failure morphology of the specimen is basically the same as that of the atmospheric temperature specimen. However, after elevated temperatures, the plastic improvement effect of crumb rubber on the specimen is significantly reduced. As the heating temperature increases, the morphology of the damaged material becomes more incomplete.

The relationship between the heating temperature of the specimen and the cube compressive strength and reduction rate is shown in Figure 13. Compared with the atmospheric temperature control group, the cube’s compressive strength of 0%, 10%, and 20% crumb rubber content at 300 °C is 104.04%, 96.13%, and 80.43% of that at atmospheric temperature, and decreases to 68.34%, 55.36%, and 54.39% of that at 500 °C, and decreases to 36.33%, 30.68%, and 28.91% of atmospheric temperature at 700 °C. The compressive strength of the control group slightly increased after the elevated temperature action at 300 °C. As the cement gel in the test piece has not yet been fully hydrated, it quickly hydrates at elevated temperatures, which improves the strength of the concrete. The experimental group with crumb rubber did not exhibit this phenomenon. The compressive strength of the specimen was reduced due to the addition of crumb rubber, and the increase in concrete strength caused by the elevated temperature was offset. At the same temperature, the reduction rate of compressive strength increases with an increase in crumb rubber content. At the same crumb rubber content, the reduction rate of compressive strength increases with an increase in heating temperature.

The internal structure of concrete is altered with crumb rubber. Due to the weak strength of crumb rubber itself, it also has the function of air entrainment during concrete mixing. At the same time, the bonding between nonpolar crumb rubber and cement matrix is weak, and there is a certain transition interface between crumb rubber and cement matrix, which has a significant impact on the strength of concrete. With the increase in crumb rubber content, the compressive strength of concrete shows a downward trend. At elevated temperatures, concrete undergoes a series of complex physicochemical reactions, causing the originally hard aggregates and cement stones to become loose. With the increase in crumb rubber content, additional, wider cracks appear on the surface of the specimen. As the weak part of the specimen, cracks penetrate under load, reducing the load-bearing capacity of the specimen.

### 3.5. Splitting Tensile Strength

At the initial loading stage of the specimen at atmospheric temperature, there was no significant change on the surface of the control group specimen. When approaching the ultimate load, cracks appear along the splitting surface and then develop into wider cracks, and the specimen is split in two. When the CRC specimen approaches the ultimate load, small cracks appear on the splitting surface, and the cracks develop slowly. During failure, the specimen was relatively intact and did not split into two halves. The failure morphology of the CRC-20 specimen is more complete than that of the CRC-10 specimen, as shown in Figure 14. With the increase in temperature, the more cracks that develop on the surface of the specimen, the looser the specimen.

With changes in crumb rubber content and temperature, the internal morphology of the split specimen shows different changes, as shown in Figure 15. The closer the specimen is to the inside, the darker the color. The reason is that an uneven temperature field is formed inside the specimen. The closer to the surface of the specimen, the larger the temperature gradient. The crumb rubber undergoes a phase change, and the surface layer of the specimen changes from solid to black-brown gas. The gas spills along the cracks on the surface, while the internal gas cannot overflow and becomes solid during the cooling process.

The relationship between the heating temperature of the specimen and the splitting tensile strength and reduction rate is shown in Figure 16. Compared with the atmospheric temperature control group, the splitting tensile strengths of the specimens with 0%, 10%, and 20% crumb rubber content decreased to 80.58%, 56.84%, and 38.46% of the atmospheric temperature at 300 °C, 27.43%, 19.84%, and 18.64% of the atmospheric temperature at 500 °C, 13.83%, 9.12%, and 7.1% of the atmospheric temperature at 700 °C, respectively. The load-bearing capacity of the specimen is completely lost at 700 °C because the crumb rubber has basic qualitative changes, completely losing the ability to delay and block the development of cracks. Compared with the control group, the splitting tensile strength decreases more obviously with the increase in crumb rubber content. Some studies show that temperature has a significant effect on the tensile strength properties of concrete, especially in the case of high-strength concrete. The rate of decrease in tensile strength is much faster beyond 300 °C [47].

For the same crumb rubber content, the reduction rate of the splitting tensile strength of the specimen is obviously greater than that of the cube compressive strength. With the increase in crumb rubber content, the reduction rate is higher than that of cube compressive strength under the same condition. There are micro-cracks in the binding part of crumb rubber and concrete, and splitting tensile is sensitive to cracks. The influence of microcracks or defects on the splitting tensile strength of concrete is greater than that of the cube compressive strength. Similar conclusions are obtained in the elevated temperature test of ordinary concrete [35]. Therefore, the influence of crumb rubber on the splitting tensile strength of concrete after elevated temperatures is greater than the cube compressive strength.

### 3.6. Axial Compressive Strength

At atmospheric temperature, small longitudinal cracks appear in the control group with an increase in load. When approaching the ultimate load, the microcracks rapidly expand, one of the longitudinal cracks develops into the main crack, and the specimen rapidly loses its load-bearing capacity. The CRC specimens developed microcracks with the increase in load. When approaching the ultimate load, the microcracks develop into a more obvious crack, and the specimen loses its load-bearing capacity while there is no longitudinally penetrating main crack. The specimen is still intact after failure, and the specimen with 20% crumb rubber content is more complete than that with 10% crumb rubber content. With the increase in crumb rubber content, the specimen shows obvious plastic failure characteristics, as shown in Figure 17. At elevated temperatures, a number of longitudinal cracks appear on the surface of the specimen. In the process of loading, cracks develop continuously, and longitudinal cracks are the main failure. The higher the temperature, the worse the integrity of the specimen after failure.

The relationship between the heating temperature of the specimen and the axial compressive strength and reduction rate is shown in Figure 18. Compared with the atmospheric temperature control group, the axial compressive strengths of the specimens with 0%, 10%, and 20% crumb rubber content decreased to 95.81%, 75.33%, and 62.81% of the atmospheric temperature at 300 °C, 73.51%, 57.37%, and 41.94% of the atmospheric temperature at 500 °C, and 32.8%, 24.92%, and 19.3% of the atmospheric temperature at 700 °C, respectively. The strength reduction rate of the control group is small after treatment at 300 °C, while the strength reduction rate of the test group is obvious after treatment at 300 °C. At 700 °C, the axial compressive strength of all specimens decreases sharply. Compared with the control group, the axial compressive strength decreases more with the increase in crumb rubber content.

### 3.7. Elastic Modulus under Static Pressure

The relationship between the heating temperature of the specimen and the elastic modulus and reduction rate is shown in Figure 19. Compared with the atmospheric temperature control group, the elastic modulus of the specimens with 0%, 10%, and 20% crumb rubber content decrease to 55.45%, 56.18%, and 48.62% of the atmospheric temperature at 300 °C, 33.53%, 32.75%, and 24.94% of the atmospheric temperature at 500 °C, and 17.15%, 13.43%, and 7.99% of the atmospheric temperature at 700 °C, respectively. At 300 °C, the elastic modulus of all specimens drop sharply. Compared with the control group, the elastic modulus decreases more with the increase in crumb rubber content.

With the same crumb rubber content and temperature, the reduction rates of cubic compressive strength, axial compressive strength, elastic modulus, and splitting tensile strength increase successively, as shown in Table 4. As can be seen from Table 4, the degree of influence of elevated temperature on the strength and elastic modulus is as follows: splitting tensile strength > elastic modulus > axial compressive strength > cubic compressive strength.

### 3.8. Uniaxial Compression Stress-Strain Relationship

This test mainly records the rising section of the stress-strain curve of crumb rubber concrete, and the stress-strain relationship of specimens with different crumb rubber content at different temperatures is shown in Figure 20.

As can be seen from Figure 20, the stress-strain rules of the three groups of specimens are similar. With the increase in temperature, the stress-strain curve of each group gradually flattens out, and the peak stress of concrete obviously decreases while the strain corresponding to the peak stress obviously increases. At the same temperature, there is a greater decrease in the peak stress with the increase in crumb rubber content, but the strain corresponding to the peak stress is basically the same, and the stress-strain curve is smoother. At the initial stage of the stress-strain curve, the specimen is in an elastic state, and the curve increases approximately linearly. While the temperature is above 500 °C, there are many cracks on the surface of the specimen, and there is low stress in the straight section. In the second stage, the slope of the stress-strain curve gradually decreases and the plastic deformation develops rapidly, showing obvious nonlinearity. Due to the continuous development of surface cracks, peak stress is greatly reduced. In the third stage, cracks on the surface of the specimen further develop, forming macroscopic oblique cracks, and the specimen loses bearing capacity. The more rubber is added, the more cracks are found on the surface, but no sudden brittle failure occurs. At 300 °C, the strain value of the control group and CRC groups is about 1.8 times the atmospheric temperature, and at 700 °C, the strain value is about 4 times the atmospheric temperature.

The measured data are analyzed by multivariate nonlinear regression using Matlab. The measured stress-strain data of concrete with three different crumb rubber contents under atmospheric temperatures of 300 °C, 500 °C, and 700 °C are separately fitted. The relationship between stress-strain and crumb rubber content under uniaxial compression is shown in Figure 21. The stress-strain fitting curves of specimens with different crumb rubber content show the fitting results are basically consistent with the test results, as shown in Figure 22.

Compared with relevant studies [50,51], it is found that at atmospheric temperature, with an increase in crumb rubber content, the axial compressive strength, elastic modulus, and flexural strength show a trend of decreasing while the deformation ability increases, which is a general rule of CRC. With the increase in temperature, the strength, elastic modulus, and thermal conductivity show a trend toward decreasing. Thus, it can be used in some application scenarios to use its deformation ability rather than strength. The replacement rate of crumb rubber in this test is between 10% and 20%; however, the replacement range is limited. The replacement rate can be extended to 50% with further research. In addition, after different elevated temperatures, changes in the internal morphology and chemical composition of crumb rubber concrete are not considered in this study and will be considered in further studies.

## 4. Conclusions

The effects of temperature and crumb rubber content on the strength, elastic modulus, and stress-strain of CRC were analyzed by testing the experimental phenomena, mass loss, heat conduction characteristics, and mechanical properties of concrete with different crumb rubber contents under elevated temperatures. Compared with concrete without crumb rubber and at atmospheric temperature, the mechanical properties of CRC at elevated temperatures were clarified. The conclusions are as follows:(1)With the increase in heating temperature, the apparent color of the specimen becomes lighter, the surface cracks of the specimen become wider, and the mass loss of the specimen increases. With the increase in crumb rubber content, the cracks on the surface of the specimen become wider, the mass loss becomes larger, and the heat conduction rate of concrete decreases.(2)An elevated temperature has a great influence on the strength and elastic modulus of CRC. With the increase in heating temperature, the cube compressive strength of the control group increases slightly at 300 °C, while the cube compressive strength of the CRC decreases, and the split tensile strength, axial compressive strength, and elastic modulus of the control group and CRC all decrease. However, the integrity of the CRC specimen after failure is better, and the plastic failure characteristics are more obvious.(3)The cubic compressive strength, splitting tensile strength, axial compressive strength, and elastic modulus decrease more obviously with the increase in crumb rubber content. The degree of influence of elevated temperature on strength and elastic modulus is as follows: splitting tensile strength > elastic modulus > axial compressive strength > cubic compressive strength.(4)With the increase in heating temperature, the stress-strain curve tends to be flat, the peak stress decreases, and the corresponding peak strain obviously increases. At the same temperature, the peak stress decreases more with the increase in crumb rubber content, but the corresponding peak strain is basically the same. After multiple nonlinear regression analyses and the fitting of stress-strain data from three kinds of crumb rubber mixtures, the stress-strain fitting curves of specimens with different crumb rubber mixtures are obtained. The fitting results are basically consistent with the experimental results.

## Figures and Tables

**Figure 1 polymers-15-03102-f001:**
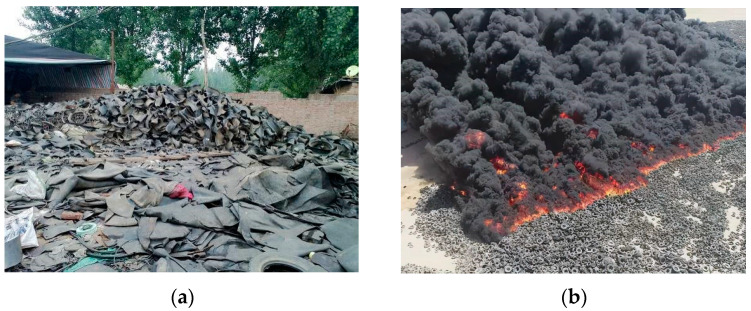
Waste rubber tires: (**a**) Waste rubber accumulation; (**b**) Burning of waste rubber tires in Kuwait.

**Figure 2 polymers-15-03102-f002:**
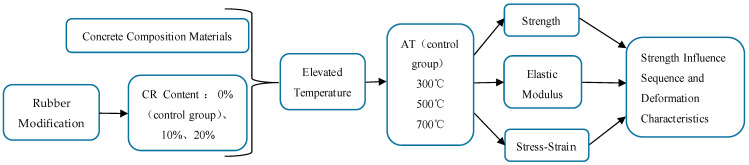
Schematic flow diagram of this study.

**Figure 3 polymers-15-03102-f003:**
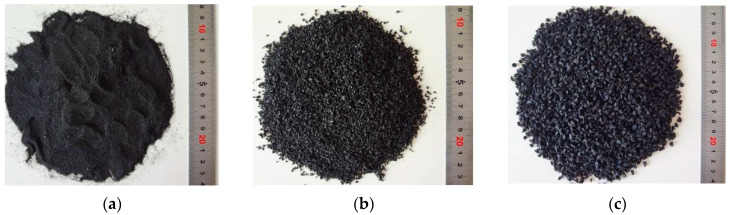
Crumb rubber: (**a**) 0.6 mm; (**b**) 1–3 mm; (**c**) 3–5 mm.

**Figure 4 polymers-15-03102-f004:**
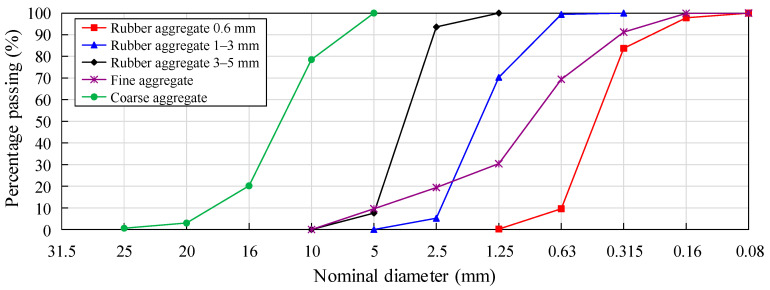
Grading curve of coarse aggregate, fine aggregate, and crumb rubber.

**Figure 5 polymers-15-03102-f005:**
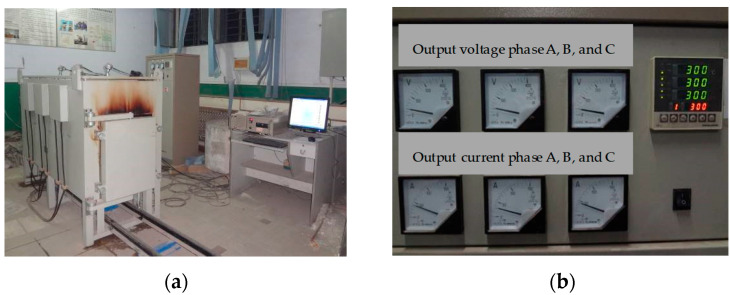
Elevated temperature testing furnace: (**a**) Elevated temperature furnace; (**b**) temperature control box.

**Figure 6 polymers-15-03102-f006:**
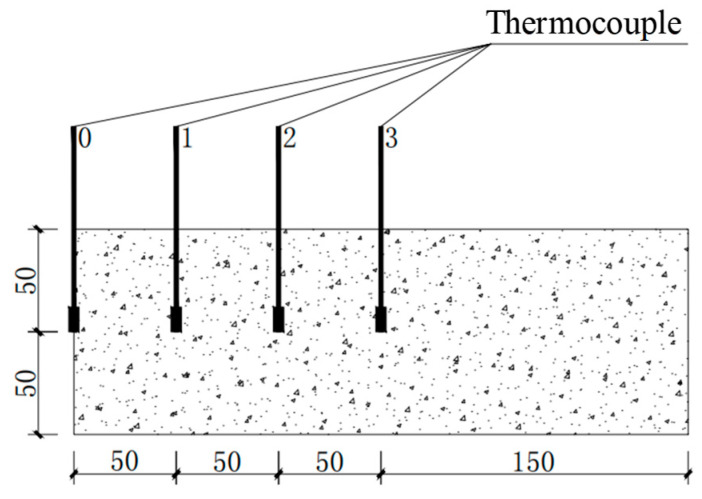
The embedding position of thermocouples.

**Figure 7 polymers-15-03102-f007:**
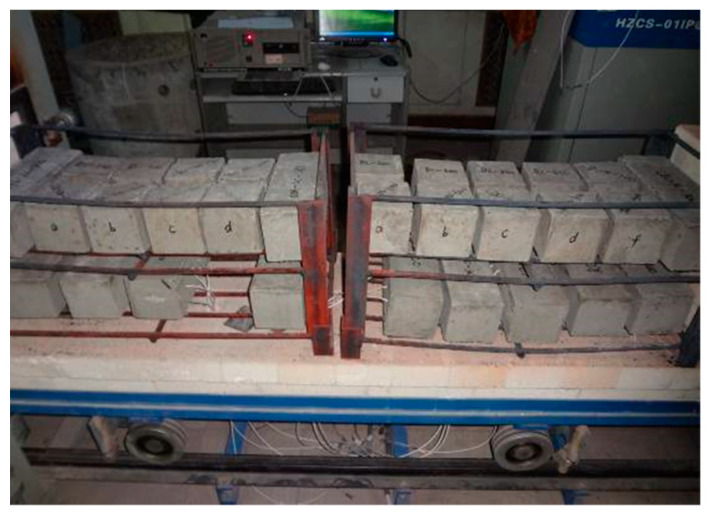
Placement of specimens.

**Figure 8 polymers-15-03102-f008:**
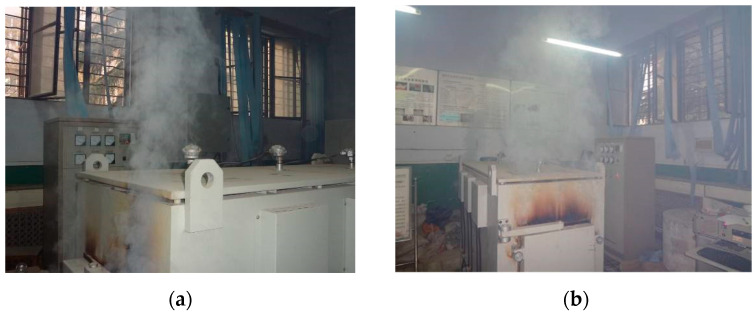
Large amounts of water vapor and pungent smoke are generated during the experiment at (**a**) 250 °C and (**b**) 400 °C.

**Figure 9 polymers-15-03102-f009:**
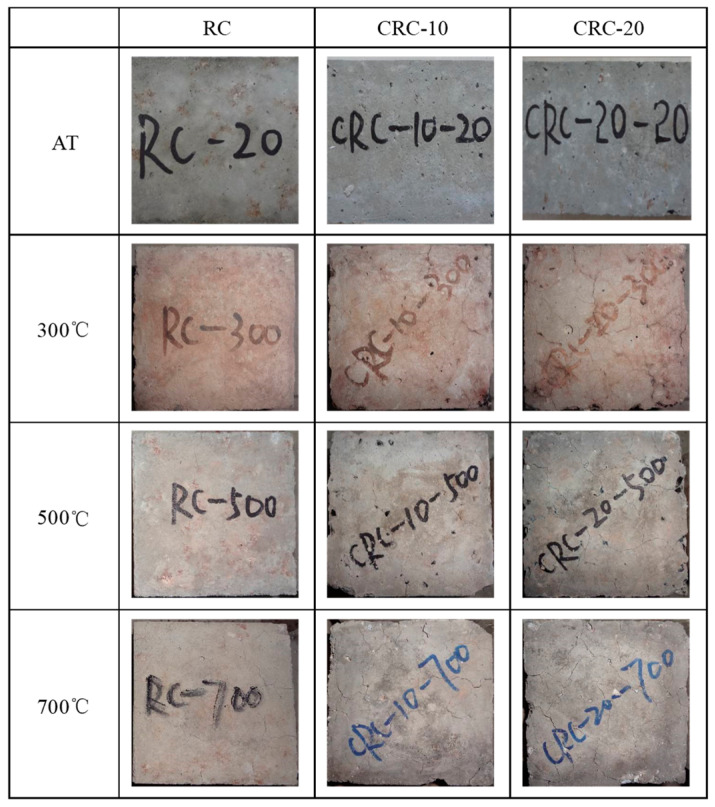
Appearance of specimens after different elevated temperatures.

**Figure 10 polymers-15-03102-f010:**
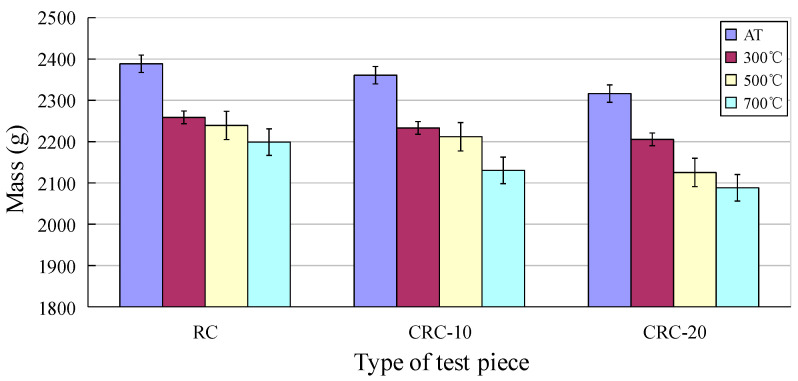
Change in specimen quality after elevated temperature.

**Figure 11 polymers-15-03102-f011:**
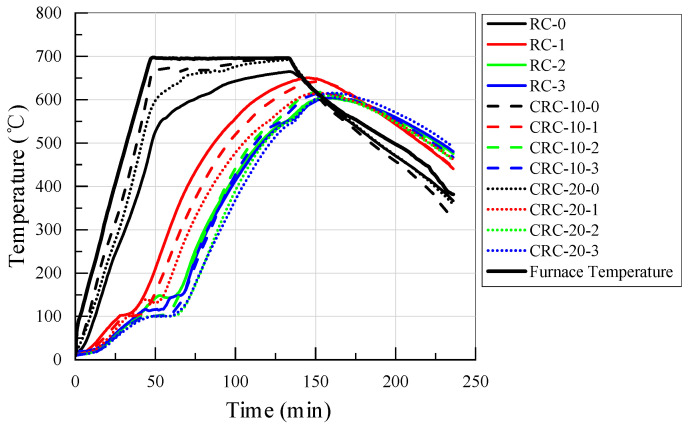
Time-temperature curves of the surface and interior of the specimen at 700 °C.

**Figure 12 polymers-15-03102-f012:**
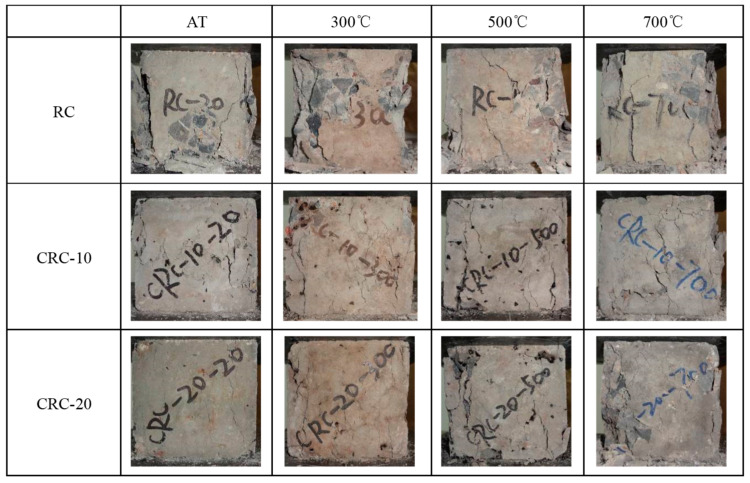
Failure morphology of concrete under different crumb rubber content and temperatures.

**Figure 13 polymers-15-03102-f013:**
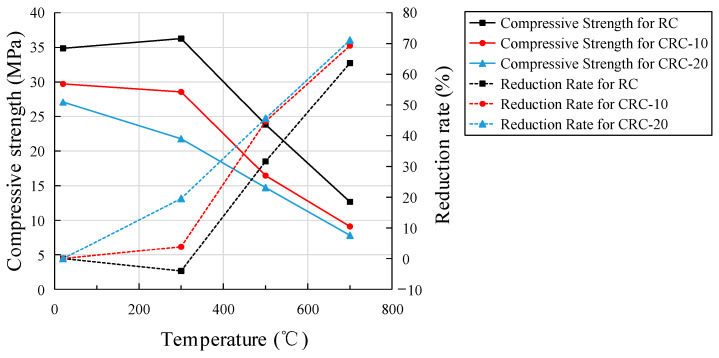
The relationship between temperature, cube compressive strength, and reduction rate.

**Figure 14 polymers-15-03102-f014:**
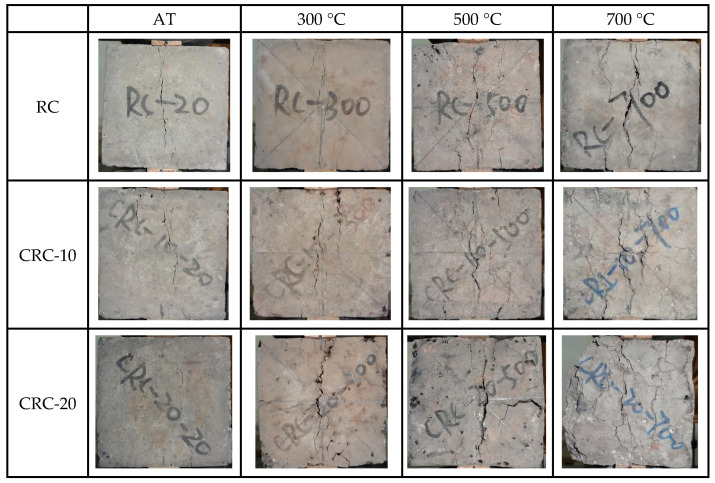
Splitting tensile failure modes of specimens under different temperatures and crumb rubber contents.

**Figure 15 polymers-15-03102-f015:**
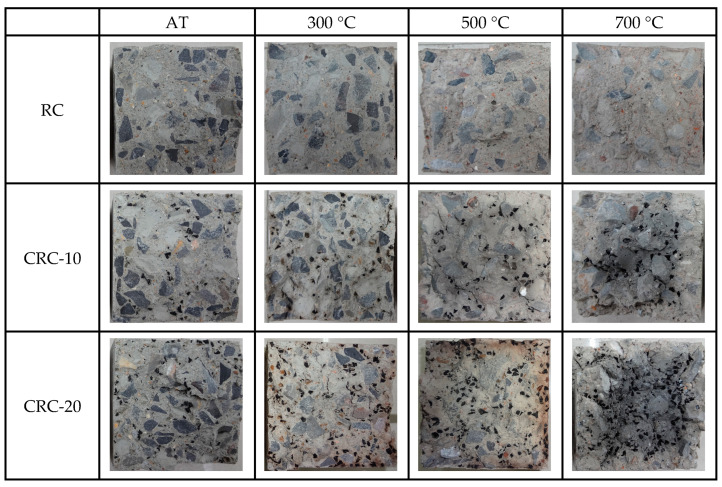
Internal morphology of specimens after splitting failure.

**Figure 16 polymers-15-03102-f016:**
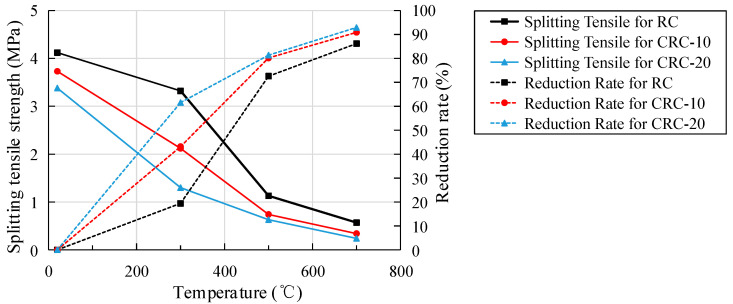
The relationship between specimen temperature and the splitting tensile strength and reduction rate.

**Figure 17 polymers-15-03102-f017:**
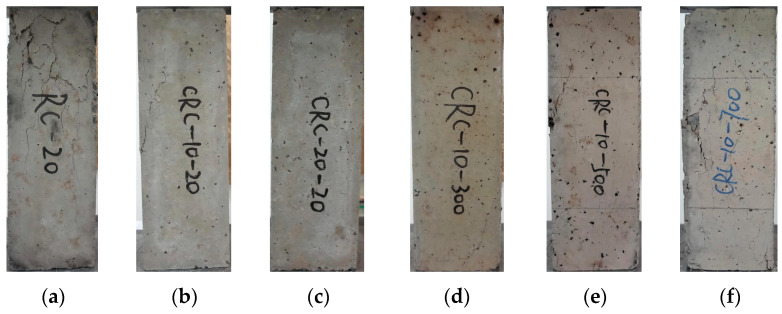
Axial failure modes of specimens under different temperatures and crumb rubber contents: (**a**) RC-20; (**b**) CRC-10-20; (**c**) CRC-20-20; (**d**) CRC-10-300; (**e**) CRC-10-500; (**f**) CRC-10-700.

**Figure 18 polymers-15-03102-f018:**
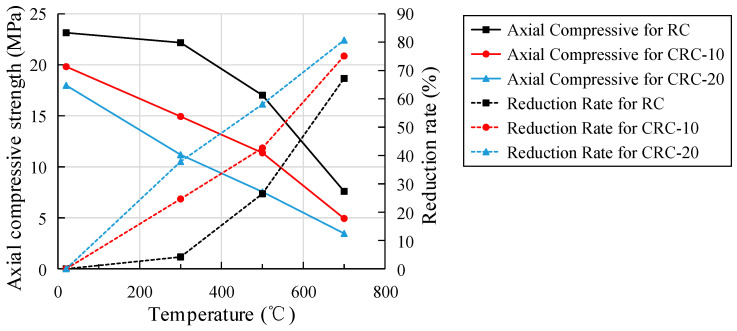
The relationship between specimen temperature and axial compressive strength and reduction rate.

**Figure 19 polymers-15-03102-f019:**
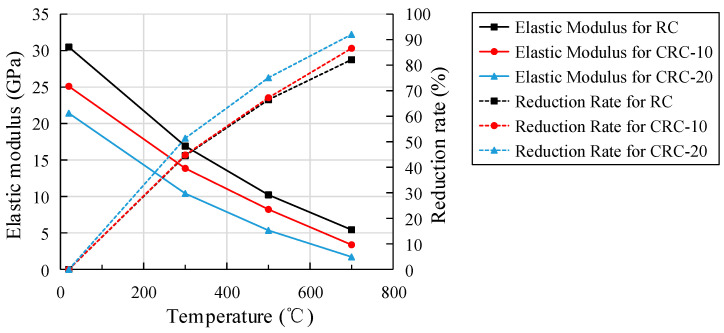
The relationship between temperature, elastic modulus, and reduction rate.

**Figure 20 polymers-15-03102-f020:**
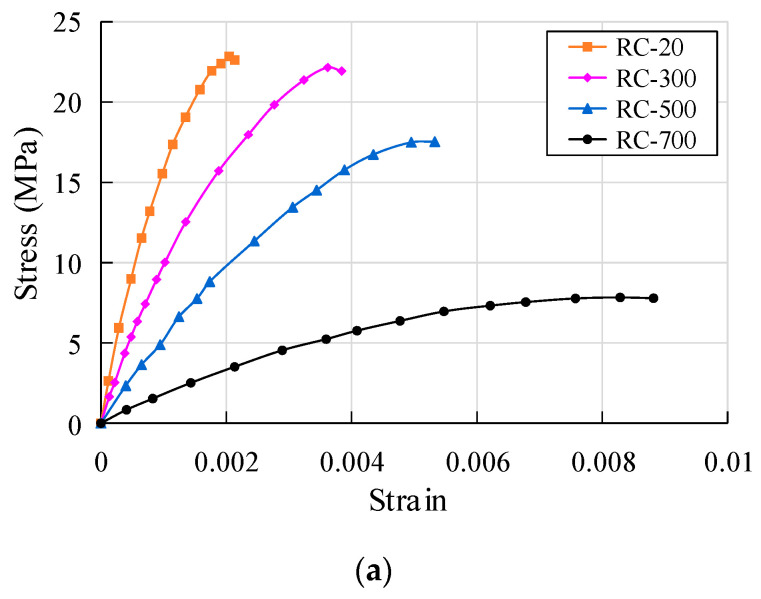
Stress-strain relationship of specimens with different crumb rubber contents under different temperatures: (**a**) RC; (**b**) CRC-10; (**c**) CRC-20.

**Figure 21 polymers-15-03102-f021:**
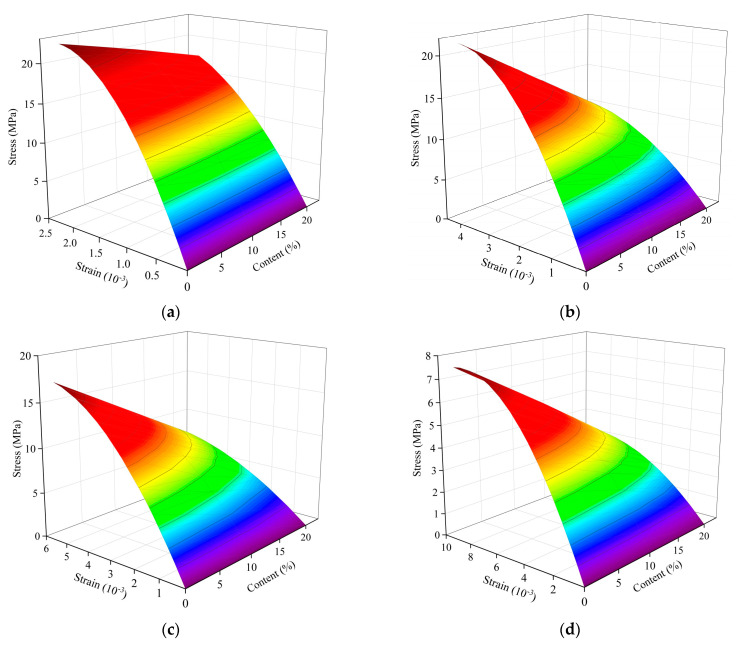
The relationship of stress-strain-crumb rubber content at different temperatures: (**a**) AT; (**b**) 300 °C; (**c**) 500 °C; (**d**) 700 °C.

**Figure 22 polymers-15-03102-f022:**
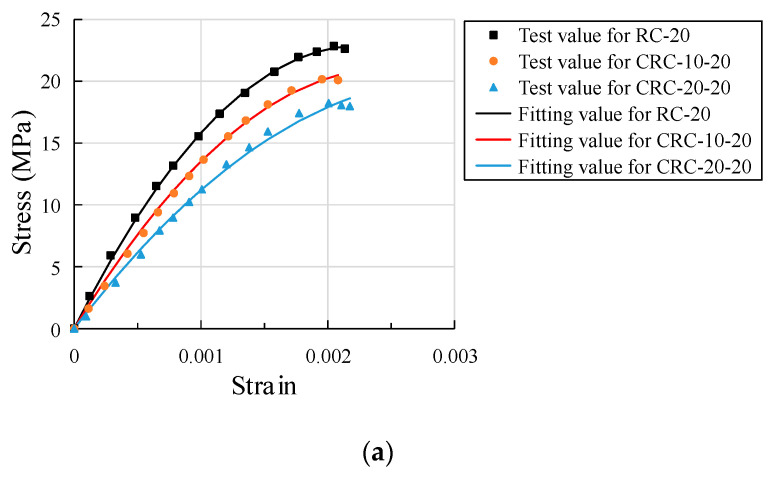
Comparison of experimental results and fitting results at different temperatures: (**a**) AT; (**b**) 300 °C; (**c**) 500 °C; (**d**) 700 °C.

**Table 1 polymers-15-03102-t001:** Physical and mechanical performance indicators of cement.

Fineness/%	Alkali Content/%	Setting Time/min		Flexural Strength/MPa		Compressive Strength/MPa	
		Initial Setting Time	Final Setting Time	3 d	28 d	3 d	28 d
<10	0.56	153	203	5.6	8.7	30	56

**Table 2 polymers-15-03102-t002:** CRC mix ratio kg/m^3^.

Specimen	Cement	Crumb Rubber	Fine Aggregate	Coarse Aggregate	Fly Ash	Water	Water-Reducing Agent	Water-Cement Ratio	Sand Content
RC	280	0	848	1037	70	175	5.7	0.5	0.45
CRC-10	280	34.7	763.2	1037	70	175	5.7	0.5	0.42
CRC-20	280	69.4	678.4	1037	70	175	5.7	0.5	0.40

**Table 3 polymers-15-03102-t003:** Effective quantity and grouping of experiments.

Specimen	Heating Temperature	Cube Compressive (100 mm × 100 mm × 100 mm)	Splitting Tensile (100 mm × 100 mm × 100 mm)	Axial Compressive (100 mm × 100 mm × 300 mm)	Elastic Modulus (100 mm × 100 mm × 300 mm)	Stress-Strain (100 mm × 100 mm × 300 mm)
RC-20	AT	3	3	3	3	3
CRC-10-20		3	3	3	3	3
CRC-20-20		3	3	3	3	3
RC-300	300 °C	3	3	3	3	3
CRC-10-300		3	3	3	3	3
CRC-20-300		3	3	3	3	3
RC-500	500 °C	3	3	3	3	3
CRC-10-500		3	3	3	3	3
CRC-20-500		3	3	3	3	3
RC-700	700 °C	3	3	3	3	3
CRC-10-700		3	3	3	3	3
CRC-20-700		3	3	3	3	3

RC: reference concrete (without crumb rubber); CRC: crumb rubber concrete; RC-Y: Y stands for temperature, which is atmospheric temperature, 300 °C, 500 °C, and 700 °C, respectively; CRC-X-Y: X stands for the crumb rubber content, which is 10% and 20%, respectively.

**Table 4 polymers-15-03102-t004:** Strength reduction rate under different crumb rubber content and different temperatures.

Type	RC				CRC-10				CRC-20			
	CCS/%	ACS/%	EM/%	STS/%	CCS/%	ACS/%	EM/%	STS/%	CCS/%	ACS/%	EM/%	STS/%
AT	0	0	0	0	0	0	0	0	0	0	0	0
300 °C	−4.04	4.19	44.55	19.42	3.87	24.67	44.82	43.16	19.57	37.82	51.38	61.54
500 °C	31.66	26.49	66.47	72.57	44.64	42.63	67.25	80.16	45.61	58.06	75.06	81.36
700 °C	63.67	67.2	82.15	86.17	69.32	75.08	86.57	90.88	71.09	80.7	92.01	92.90

Cubic compressive strength: CCS; Axial compressive strength: ACS; Elastic modulus: EM; Splitting tensile strength: STS.

## Data Availability

The data used to support the findings of this study are included within the article.

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
