# Peer review of "Experimental Study on the Mechanical Properties of Crumb Rubber Concrete after Elevated Temperature"

_polymers, 2023, doi:10.3390/polym15143102_

Round 1

Reviewer 1 Report

Experimental study on the mechanical properties of crumb rubber concrete after elevated temperature

Manuscript Number:

In the present paper authors provide an experimental investigation on the effect of high elevated temperature on the Rubberized concrete.  However, the paper requires some minor improvement before it can be recommended for publication, it is proposed to re-submit a thoroughly revised version of the manuscript, considering the following comments.

1.     Overall recommendation should be reported in one sentence at the end of the abstract 2.     The authors should overview the recent progress made in the relevant area in the past two years or so. Such as https://doi.org/10.1007/s43452-022-00464-y; https://doi.org/10.1016/j.istruc.2022.10.049.....etc 3.     The introduction needs to be improved. The discussion of results need to be enhanced by comparing with the other researchers 4.     Author should highlight the assumptions and limitations and future research direction of the study. 5.     The paper is well written and it is easy to follow, only the authors needs to go thoroughly revised version to correct the typo-mistake

6.     Add some of the important findings to conclusion

Reviewer 2 Report

Compare your investigations also with other papers

https://doi.org/10.1016/j.cscm.2018.e00184

https://doi.org/10.1016/j.rser.2015.10.092

https://doi.org/10.1016/j.conbuildmat.2012.04.068

Please correct typing errors

490 ofCRC

478, 505 stress – strain

Reviewer 3 Report

The paper is neat and clearly written and the authors corellate in an efficient way all their data. For the further improvement of the manuscript, I would recommend some modifications/additional information in the materials morphological evaluation. Although the authors discuss cracks andd appearance of the samples following their testing, there is no actual morphological analysis of the fracture cross section and/or the surface modifications after mechanical an/or thermal testing. Figure 15 presents the "internal morphology" of the samples, but it is only a visual analysis. In order to have a complete image of the samples modifications after the tests they were subjected to, I would recommend performing some scanning electron microscopy investigations on the samples, or at least some optical microscopy analysis that would allow an actual magnification of the visualized area.

There are no significant English language flaws in the manuscript.

Reviewer 4 Report

Polymers

Experimental study on the mechanical properties of crumb rubber concrete after elevated temperature

Comments:

This study investigated the effects of elevated temperature on the mass loss, thermal conductivity characteristics, compressive strength, splitting tensile strength, axial comprehensive strength, elastic modulus and stress-strain characteristics of crumb rubber concrete. It is interesting and valuable for practical engineering and circular economy. Here are some comments/suggestions for further improving the manuscript.

* The abstract is too long, and it looks like conclusions. So, please make it more concise.

* Figure 2 should be re-organized with clearer research objective.

* The high-temperature would lead to the degradation of crumb rubber, and thus the results should be further discussed and explained with that point, which can refer to Investigating the role of swelling-degradation degree of crumb rubber on CR/SBS modified porous asphalt binder and mixture. Construction and Building Materials. 2021, 124048. Extruded tire crumb-rubber recycled polyethylene melt blend as asphalt composite additive for enhancing the performance of binder. Journal of Materials in Civil Engineering. 2020, 04019373.

* Table 1: chemical components of cement and crumb rubber are expected.

* Figure 3: the unit of crumb rubber particle size should be the same (select one from mesh and mm).

* More quantitative conclusions are expected.

* Some recommendations for future work can be supplemented.

* The English editing of the whole manuscript has to be improved carefully.

The English editing of the whole manuscript has to be improved carefully.

Reviewer 5 Report

The submitted Article with the Manuscript ID: polymers-2441579 and the Title: “Experimental study on the mechanical properties of crumb rubber concrete after elevated temperature” investigates the effects of crumb rubber content and heating temperatures on the strength, elastic modulus, and stress–strain of crumb rubber concrete (CRC). It focuses primarily on inquiring about the CRC's failure mechanism after loading with elevated temperature as a reference basis for the fire resistance design, fire resistance limit analysis, and residual bearing capacity calculation of CRC structures after the fire. The paper has a satisfactory scientific approach to the subject. However, some issues, questions, and clarifications should be amended, and the paper needs some improvement to reach the required scientific level. The following comments and suggestions are raised for the authors' reference:

  1. In Figure 2, the Schematic flow diagram of the study maybe should be moved to the section "2. Materials and Methods" instead of the section "1. Introduction" since there is a specific section targeting the methodology of the study.
  2. The literature background provided in section "1. Introduction" is limited. Therefore, it is necessary to state what has been done and what has not been done in the specific area of the investigation of the mechanical properties of CRC, also focusing on the non-destructive evaluation techniques used for full-scale monitoring. Since the study is trying to evaluate the mechanical behavior of CRC after loading with elevated temperatures, a literature overview for fire damage in concrete elements could also be helpful. Moreover, supplementary literature is needed utilizing fracture characteristics of the microstructure of cementitious specimens after mechanical testing, which is of great importance for understanding the crack path during the distribution of damage in the fracture surface, also focusing on specimens that have been treated in high temperatures. In this direction, the following studies are some examples of relatively published articles that are suggested to be considered (order by date):

- “Acoustic emission of fire damaged fiber reinforced concrete”, SPIE, Smart Materials and Nondestructive Evaluation for Energy Systems, 2016.

- “Cracking diagnosis in fiber-reinforced concrete with synthetic fibers using piezoelectric transducers”, Fibers, 2022.

- “Cement mortar containing crumb rubber coated with geopolymer: From microstructural properties to compressive strength”, Construction and Building Materials, 2023.

  1. In line 141, are the substitution rates % per mass or volume?
  2. In lines 146-147, what is the speed value for mixing the crumb rubber and fine aggregates?
  3. What is the sequence of adding the crumb rubber and sand into the mixture for producing the CRC? A better description of the mixture procedure is needed.
  4. In line 153, a photo or schematic representation or a more detailed description may be useful to understand the procedure of manual vibration with two loadings and vibration steps.
  1. Why has the heating rate of 150C/min been selected? Maybe this heating rate follows some standards and some limitations?
  2. Why has not been adopted a more uniform controlled cooling procedure? Maybe this sudden loss of temperature affects the mechanical properties of the specimens, especially the surface integrity and microcracking, and how could be assumed that all specimens have the same cooling effect since their location from the door of the furnace is different from each other?
  3. In Table 3, a description of the code name of specimens could be helpful.
  4. How can be explained that none of the specimens have shown any spalling phenomenon? Not even the control specimens with 100% natural aggregates? An explanation is needed.
  5. In Figure 9, an explanation is needed why more cracks and longer crack propagation is being seen on CRC specimens.
  6. In Figure 11, an explanation of the decrease of thermal conductivity in elevated temperatures is needed.
  7. Why is the surface temperature of the CRC specimens higher than the control specimen respectively?
  8. In Figure 13, why the reduction rate of compressive strength (%) in elevated temperatures is higher for the crumb rubber content CRC-20 compared to the control specimens?
  9. In the caption of Figure 14, where is the CRC- 20 accordingly?
  10. In lines 371-372, an explanation is needed why the reduction rate of the splitting tensile strength of the specimen is greater than that of the cube compressive strength for the same crumb rubber content.

Round 2

Reviewer 5 Report

The revised Article with the Manuscript ID: polymers-2441579-v2 and the Title: “Experimental study on the mechanical properties of crumb rubber concrete after elevated temperature” has been improved. The efforts performed by the authors to consider the recommendations and to respond to all the criticisms of the previous review comments are appreciated. Most of them have been respond adequately. The revised article has adequate novelty and an acceptable scientific approach. The manuscript is well-structured and easily understood. Furthermore, the manuscript is well-revised; therefore, this new version of the paper is suggested to be accepted for publication without further re-review after addressing the following minor revisions:

  1. In Fig. 11, the temperature plateau at 100℃ presented in specimens RC is suggested to be discussed. Some comments derived from refs [48, 49] would be useful to be added in this study, too.
  2. The non-destructive evaluation techniques to observe concrete's internal cracking diagnosis and failure could be expanded to modern methods using smart materials and efficient early prognosis techniques.
  3. Concerning the surface temperature of the CRC specimens that is higher than the control specimen, it is suggested to add some comments about the crumb rubber on the surface of CRC that was melted and burned, and the lower thermal conductivity of CRC.
